# Needs assessment of a pythiosis continuing professional development program

**Surachai Leksuwankun[1], Rongpong Plongla[1]\*, Nathanich Eamrurksiri[2], Pattama Torvorapanit[1,3], Kasidis Phongkhun[1], Nattapong Langsiri[4], Tanaporn Meejun[5], Karan Srisurapanont[5], Jaedvara Thanakitcharu[6], Bhoowit Lerttiendamrong[7], Achitpol Thongkam[4], Kasama Manothummetha[8], Nipat Chuleerarux[9], Chatphatai Moonla[1,10], Navaporn Worasilchai[11,12], Ariya Chindamporn[4], Nitipong Permpalung[4,8], Saman Nematollahi[13]**

**1** Department of Medicine, Faculty of Medicine, Chulalongkorn University and King Chulalongkorn Memorial Hospital, Bangkok, Thailand, **2** Department of Surgery, Faculty of Medicine, Chulalongkorn University and King Chulalongkorn Memorial Hospital, Bangkok, Thailand, **3** Thai Red Cross Emerging Infectious Diseases Clinical Center, King Chulalongkorn Memorial Hospital, Bangkok, Thailand, **4** Department of Microbiology, Faculty of Medicine, Chulalongkorn University, Bangkok, Thailand, **5** Faculty of Medicine, Chiang Mai University, Chiang Mai, Thailand, **6** Panyananthaphikkhu Chonprathan Medical Center, Srinakharinwirot University, Nonthaburi, Thailand, **7** Faculty of Medicine, Chulalongkorn University, Bangkok, Thailand, **8** Department of Medicine, Johns Hopkins University School of Medicine, Baltimore, Maryland, United States of America, **9** Department of Medicine, University of Miami/Jackson Memorial Hospital, Miami, Florida, United States of America, **10** Center of Excellence in Translational Hematology, Faculty of Medicine, Chulalongkorn University, Bangkok, Thailand, **11** Faculty of Allied Health Sciences, Chulalongkorn University, Bangkok, Thailand, **12** Research Unit of Medical Mycology Diagnosis, Chulalongkorn University, Bangkok, Thailand, **13** Department of Medicine, University of Arizona College of Medicine, Tucson, Arizona, United States of America

\* rongpong.p@chula.ac.th

**Data Availability Statement:** Data presented in this study is available and freely accessible from the Supporting information.

## Abstract

### Background

Pythiosis is a rare disease with high mortality, with over 94% of cases reported from Thailand and India. Prompt diagnosis and surgery improves patient outcomes. Therefore, continuing professional development (CPD) is essential for early recognition. However, a needs assessment related to a pythiosis CPD program has not been performed.

### Objectives

We conducted a needs assessment to develop a pythiosis CPD program.

### Patients/Methods

We conducted a survey study with 267 King Chulalongkorn Memorial Hospital residents (141 internal medicine (IM) residents and 126 surgery residents). A 30-item survey consisting of a knowledge assessment, demographic section, and an attitudes portion was distributed both electronically and via paper. The data was analyzed with descriptive and inferential statistics.

**Funding:** PT received the fund for this work. This work was supported by the Health Systems Research Institute, the Ministry of Public Health, Thailand [HSRI 65-081 to PT]. The funder had no role in study design, data collection and analysis, decision to publish, or preparation of the manuscript. The website of funder was http://www.hsri.or.th.

**Competing interests:** The authors have declared that no competing interests exist.

## Results

Sixty-seven percent completed the survey (110/141 IM residents, 70/126 surgery residents). The mean score [95% confidence interval] on the knowledge assessment was 41.67% [39.64%-43.69%] across all objectives. The three domains with the highest scores were pythiosis risk factors (67.22% correct), microbiologic characteristics (50.83%), and radiographic interpretation (50.56%). The three domains with the lowest scores were laboratory investigation (15.00%), epidemiology (29.17%), and symptomatology (30.83%). Most participants noted that the program should be online with both synchronous and asynchronous sessions, with a preferred length of 60–90 minutes per session.

## Conclusion

The pythiosis CPD program should emphasize education regarding symptomatology, laboratory investigation, and epidemiology, all of which are critical for the early detection of pythiosis to decrease mortality from this devastating disease. Most respondents felt this program was necessary and should be implemented in a virtual blended format.

### Author summary

Researchers conducted a needs assessment to develop a Continuing Professional Development (CPD) program on pythiosis, a rare and often fatal disease prevalent in Thailand and India. The study surveyed 267 residents at King Chulalongkorn Memorial Hospital, revealing a 67% completion rate. The residents exhibited a mean knowledge score of 41.67%, with the highest proficiency in pythiosis risk factors (67.22%), microbiologic characteristics (50.83%), and radiographic interpretation (50.56%). Conversely, lower scores were observed in laboratory investigation (15.00%), epidemiology (29.17%), and symptomatology (30.83%). Participants expressed a preference for an online CPD program with both synchronous and asynchronous sessions, each lasting 60–90 minutes. The study emphasizes the need for education on symptomatology, laboratory investigation, and epidemiology to enhance early pythiosis detection and reduce mortality. Respondents widely supported the implementation of the proposed virtual blended CPD program.

## Introduction

Pythiosis is a rare emerging disease with high morbidity and mortality [1]. The causative pathogen is *Pythium insidiosum*, an aquatic fungal-like microorganism that lives in soil, wet environments, and agricultural lands [2]. *P. insidiosum* can infect humans via direct exposure to the pathogenic zoospores [3,4]. This low-prevalence disease is reported in tropical, subtropical, and temperate countries such as Thailand, India, the United States (US), and Brazil. Strikingly, more than 94% of human pythiosis cases are reported in India and Thailand [4]. In addition, the clinical presentation of human pythiosis is varied and can be classified into vascular, disseminated, ocular, and cutaneous/subcutaneous forms. Vascular pythiosis is the most common form in Thailand, and most of the global vascular pythiosis cases are reported from Thailand [1]. From our preliminary randomized clinical trial on vascular pythiosis treatment, we have enrolled nearly 30 cases a year, making this an underreported disease.

The current mainstay of treatment for vascular pythiosis is a surgical intervention to achieve source control. Moreover, adjunctive agents with antifungal drugs, antibiotics, and immunotherapy may improve outcomes [5–7]. Although surgery combined with medication is used as a therapeutic intervention, the outcomes are varied depending on surgical free margins, presence of arterial involvement, time to diagnosis, and time to surgery [6,8]. Therefore, early diagnosis is critical in improving morbidity and mortality [7]. However, human pythiosis is an uncommon disease and difficult to diagnose due to similar clinical presentations to other fungal infections such as mucormycosis, talaromycosis, and aspergillosis [9], in addition to other factors such as the gradual onset of vascular invasion, patient and physician under-recognition, and disease rarity. To help improve time to diagnosis, there is a need to increase awareness and knowledge of vascular pythiosis through an educational intervention [1].

Continuing professional development (CPD) is a method to update and maintain physicians' performance [10]. CPD is an effective way to improve professional practices and healthcare outcomes [11,12]. CPD methods for content delivery may be various such as formal learning, academic conferences, reading articles, and mentoring [13]. Moreover, best practices in CPD development are guided by the acronym "CRISIS", which consists of convenience, relevance, individualization, self-assessment, interest, speculation and systematic [14].

We are addressing two gaps in the literature. First, there is no current evidence on the knowledge assessment of vascular pythiosis among physicians. Second, there is limited evidence of physicians' preference for a pythiosis CPD program. Therefore, this study aims to conduct a needs assessment by assessing the knowledge gap in Thai physicians on vascular pythiosis and surveying the attitude toward CPD preference according to the CRISIS acronym.

## Methods

### Ethics statement

The authors confirmed that the ethical policies of the journal, as noted on the journal's author guidelines page, have been adhered to and the appropriate ethical review committee approval has been received from the Institutional Review Board of the Faculty of Medicine, Chulalongkorn University (certification of approval number 1734/2022). This study was conducted with implied consent by action when the participants answered the questionnaire. Therefore, formal written consent was not obtained due to anonymity.

### Educational theory and conceptual framework

The theoretical framework of this study follows the foundation for developing effective CPD from continuing medical education [10]. CPD aims to maintain and improve the clinical performance of physicians through various methods, such as training courses, reflections from clinical practice, clinical meetings, and mentoring [15]. Similarly, this study on vascular pythiosis has the ultimate goal of improving the clinical performance of physicians and improving the morbidity and mortality of vascular pythiosis. The approach to CPD planning considers various learning styles and needs, in addition to the needs assessment, educational objectives and content, selection of content delivery (such as time and place), interactivity and relevance, and self-directed learning [10,15,16] (Fig 1).

### Research aims and design

There were two research aims in this study. The first research aim was to conduct the pythiosis needs assessment and define the educational objectives and content for a pythiosis CPD

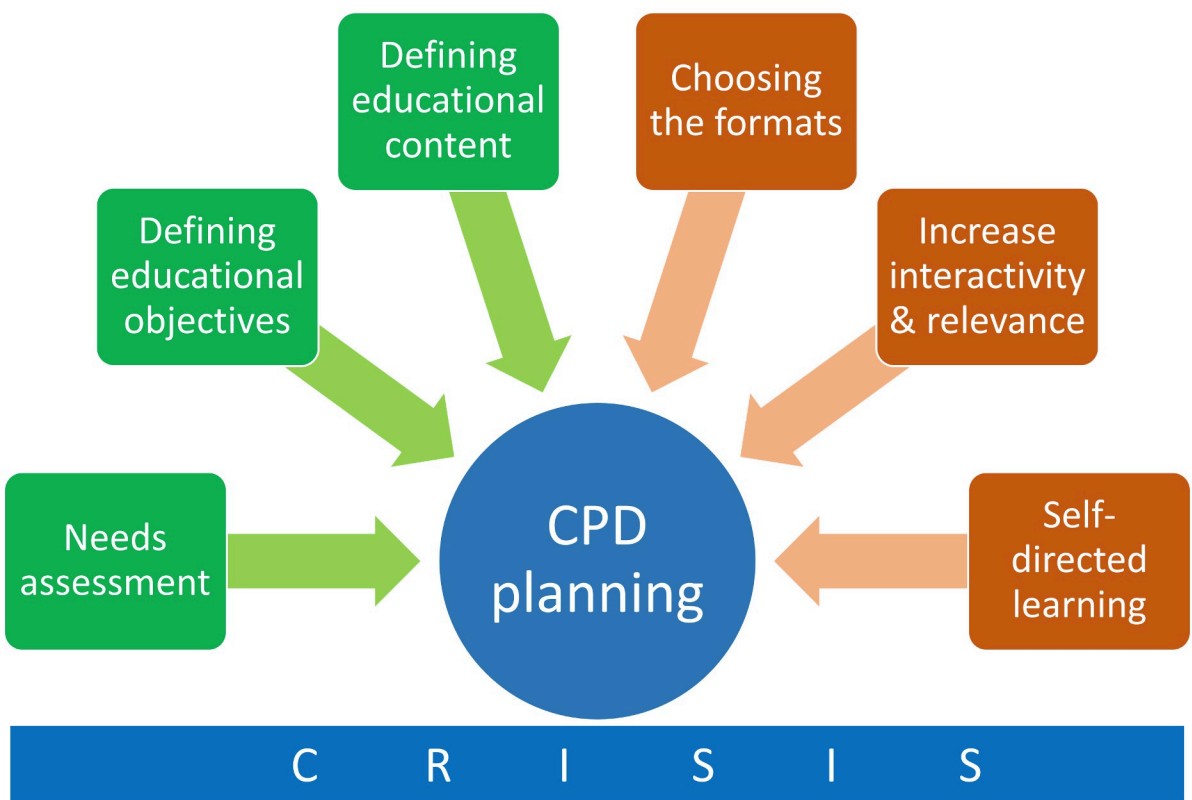

**Fig 1. Conceptual framework for CPD planning using CRISIS.**

program. The second research aim was to determine the preferred format of delivering content, how to increase interactivity and relevance, and how to implement self-directed learning. We collected data for both aims using a paper and online survey that included a knowledge assessment and questions about physician preference for CPD following the CRISIS acronym. The cross-sectional survey study was conducted between January–February 2023.

## Instrumentation

We created the survey by following survey development guidelines [17]. The questionnaire included three parts: demographic data, knowledge assessment, and an attitude portion. Demographic data included gender and the professional status of participants such as post-graduate year (PGY) and type of residency training program (Supplement 1). The test items were constructed by six experts in vascular pythiosis who are members of the Mycology, Epidemiology, and Medical Education Research Group (MERG). Each test item was mapped to one of nine learning objectives (Supplement 2). All 20 items were multiple choice questions with one single best answer. The attitudes questions followed the CRISIS acronym. First, convenience of learning was considered synchronous or asynchronous, and online or onsite learning. Second, relevance was addressed by asking about the timing of their first encounter with vascular pythiosis. Third, individualization included the amount of time and mode of learning that they would like to spend in the CPD course. Fourth, self-assessment was performed via the knowledge portion. Fifth, interest was addressed by asking about their interests in participating in a vascular pythiosis training course. Sixth, speculation and systematic covered nine important learning objectives of vascular pythiosis.

The questionnaire was reviewed separately for content validity by two experts in medical education and two experts in infectious diseases at the Faculty of Medicine, Chulalongkorn University. The questionnaire was refined after one-on-one cognitive interviews with five participants from MERG. These participants included an internal medicine resident in the U.S., two interns, and two medical students in Thailand. Additionally, the questionnaire was piloted with a cohort of 10 residents who are not internal medicine and surgery residents.

### Population, sampling, and statistics

The population in this study was physicians who encounter vascular pythiosis at the Faculty of Medicine, Chulalongkorn University. Therefore, this study included internal medicine and surgery residents. The internal medicine residency is a three-year program, and the surgery residency is a five-year program with a lower number of residents in the fourth and fifth years. The sampling technique was voluntary sampling. The sample size calculation was based on the primary research question which was to assess the needs of the Pythiosis course. This calculation followed the estimation of the means equation which consisted of 0.05 alpha and 80% power. Data from the pilot study with 10 residents revealed a 9/20 points mean score with 1.9 points of standard deviation and the anticipation of participants score was 10/20 points. The minimal sample size was 28 participants.

The statistical analysis was performed using Microsoft Excel and Stata version 17.0 (Stata-Corp, College Station, TX). The analytical process concealed the identity of all participants to keep the data anonymous. For descriptive statistics, categorical data were presented as frequency and percentage, while continuous variables were presented as mean (M) and standard deviation (SD). For inferential statistics, the estimation of the population mean was presented with 95% confidence interval (CI). Additionally, the mean scores of the internal medicine and surgery residents were compared using independent t-test and Cohens' d (d) for effect size estimation. A p-value less than 0.05 was considered statistically significant.

## Results

### Participants

The overall voluntary response was 180/267 participants (67.42%) including 110/141 (78.01%) of internal medicine residents and 70/126 (55.56%) of surgery residents. From the overall responses (N = 180), 66.11% were completed online, and 53.93% identified as male. Moreover, the demographic data revealed 32.96% PGY1, 33.52% PGY2, 24.58% PGY3, 7.26% PGY4, and 1.68% PGY5 as shown in Table 1.

Table 1. Demographic data of survey respondents.

| | | Internal medicine (n = 110)* | | Surgery (n = 70) | | Total (n = 180)* | |
|---|---|---|---|---|---|---|---|
| Gender | Male | 57 | (52.78%) | 39 | (55.71%) | 96 | (53.93%) |
| | Female | 51 | (47.22%) | 31 | (44.29%) | 82 | (46.07%) |
| PGY | PGY1 | 40 | (36.70%) | 19 | (27.14%) | 59 | (32.96%) |
| | PGY2 | 36 | (33.03%) | 24 | (34.29%) | 60 | (33.52%) |
| | PGY3 | 33 | (30.28%) | 11 | (15.71%) | 44 | (24.58%) |
| | PGY4 | 0 | (0%) | 13 | (18.57%) | 13 | (7.26%) |
| | PGY5 | 0 | (0%) | 3 | (4.29%) | 3 | (1.68%) |

PGY: Post-graduate year;

*Few participants did not complete all the questions.

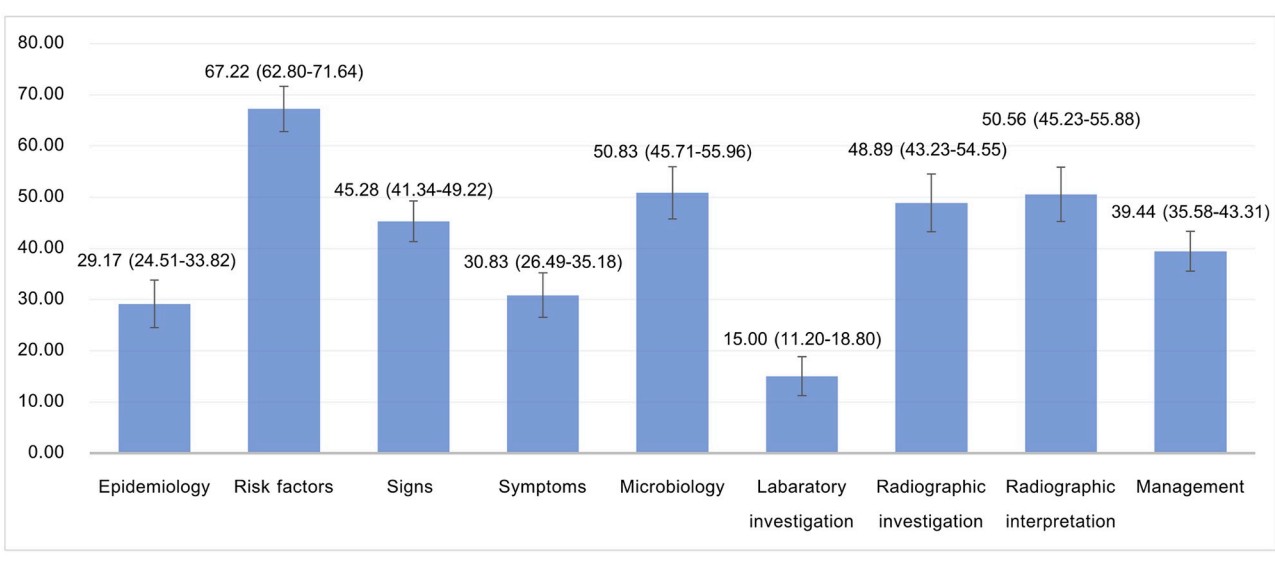

**Fig 2. Percent correct across the learning objectives with 95% confidence interval.**

## Knowledge assessment

The overall mean score [95% CI] was 41.67% [39.64%-43.69%]. The mean score was 44.14% [41.74%-46.54%] among IM residents and 37.79% [34.31%-41.26%] for surgery residents, with the difference being statistically significant (t = 3.08, $p$ = 0.0024, $d$ = 0.47).

Among the nine learning objectives, the percent correct ranged between 15.00–67.22%. The learning objectives with the highest percent correct were risk factors (M = 67.22% [62.80%-71.64%]), microbiology (M = 50.83%, [45.71%-55.96%]), and radiographic interpretation (M = 50.56%, [45.23%-55.88%]). On the other hand, the learning objectives with the lowest percent correct were laboratory investigation (M = 15.00%, [11.20%-18.80%]), epidemiology (M = 29.17%, [24.51%-33.82%]), and symptoms of vascular pythiosis (M = 30.83%, [26.49%-35.18%]) (Fig 2).

In the subgroup analysis as shown in Fig 3, the highest percent correct among internal medicine residents was risk factors (M = 73.64%, [67.95%-79.32%]), radiographic investigation (M = 54.09%, [46.84%-61.35%]), and microbiology (M = 51.36%, [45.03%-57.69%]), whereas the lowest percent correct was laboratory investigation (M = 16.82%, [11.50%-22.14%]), epidemiology (M = 27.27%, [21.32%-33.23%]), and symptoms of vascular pythiosis (M = 34.09%, [28.52%-39.66%]). Furthermore, the highest percent correct among surgery residents was risk factors (M = 57.14%, [50.63%-63.65%]), radiographic interpretation (M = 50.00%, [40.92%-59.08%]), and microbiology (M = 50.00%, [41.15%-58.85%]), while the lowest correction percentage was laboratory investigation (M = 12.14%, [6.99%-17.29%]), symptoms of vascular pythiosis (M = 25.71%, [18.76%-32.67%]), and epidemiology (M = 32.14%, [24.54%-39.75%]). Additionally, the mean score among internal medicine residents was significantly higher than surgery residents in risk factors (t = 3.72, p = 0.0003, d = 0.57), signs of vascular pythiosis (t = 2.43, p = 0.0163, d = 0.37), and radiographic investigation (t = 2.30, p = 0.0225, d = 0.35).

## Attitude toward continuing professional development

Among the survey respondents, 48.04% preferred a learning environment with both synchronous and asynchronous sessions, 35.75% preferred asynchronous learning only, and 16.20% preferred synchronous learning only (Table 2). Furthermore, 45.81% favored online platforms,

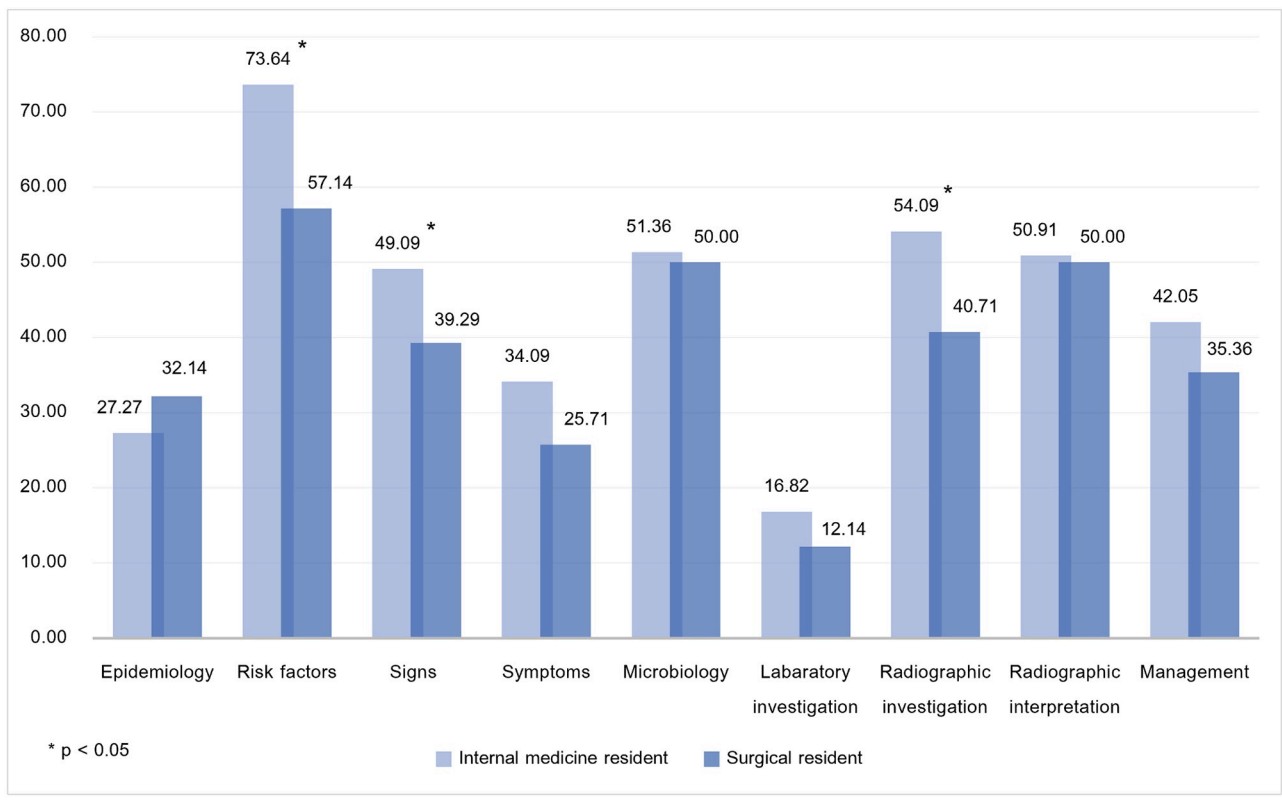

**Fig 3. Percent correct across the learning objectives, categorized by residency program.**

38.55% favored hybrid, and 15.64% favored on-site learning (Table 2). Surprisingly, 32.40% of participants had never learned about pythiosis while some of them heard about pythiosis during medical school (30.17%), residency (25.70%), internship (10.06%), or before medical school (1.68%) (Table 2). Moreover, 94.41% agreed that the vascular pythiosis course was important and 63.13% wished to participate in the course (Table 2). Additionally, 94.94% suggested that the whole course duration should not exceed 90 minutes (Table 2).

## Discussion

This study aimed to establish the needs assessment for a pythiosis CPD program. To our knowledge, this is the first educational program focused on a rare infectious disease in Thailand. Among IM and surgery residents at a single institution in Thailand, the learning objectives with the highest percent correct were risk factors, microbiology, and radiographic interpretation, whereas those with the lowest percent correct were laboratory investigation, epidemiology, and symptoms of vascular pythiosis. Moreover, the residents prefer online training courses of CPD with a blend of synchronous and asynchronous sessions. Moreover, vascular pythiosis was relevant to most residents but 32% of residents had never learned about pythiosis. Most residents were aware of the importance of this knowledge gap and wanted to participate in the training course.

Learning needs assessment is an early crucial step in developing education courses in CPD by using objective tests and attitudes toward learners' preferences [13,18–20]. There is limited evidence in learning needs assessment of infectious disease training courses, but it

**Table 2. Attitude toward continuing professional development.**

| | | Internal medicine (n = 110)* | | Surgery (n = 70) * | | Total* | |
|---|---|---|---|---|---|---|---|
| **Learning mode** | **Synchronous** | 21 | (19.27%) | 8 | (11.43%) | 29 | (16.20%) |
| | **Asynchronous** | 44 | (40.37%) | 20 | (28.57%) | 64 | (35.75%) |
| | **Synchronous and asynchronous** | 44 | (40.37%) | 42 | (60.00%) | 86 | (48.04%) |
| **Learning place** | **Onsite** | 11 | (10.00%) | 17 | (24.64%) | 28 | (15.64%) |
| | **Online** | 56 | (50.91%) | 26 | (37.68%) | 82 | (45.81%) |
| | **Hybrid** | 43 | (39.09%) | 26 | (37.68%) | 69 | (38.55%) |
| **Relevance** | **Known before medical student** | 3 | (2.73%) | 0 | (0.00%) | 3 | (1.68%) |
| | **Known during medical student** | 38 | (34.55%) | 16 | (23.19%) | 54 | (30.17%) |
| | **Known during internship** | 13 | (11.82%) | 5 | (7.25%) | 18 | (10.06%) |
| | **Known during residents** | 27 | (24.55%) | 19 | (27.54%) | 46 | (25.70%) |
| | **Never known before** | 29 | (26.36%) | 29 | (42.03%) | 58 | (32.40%) |
| **Necessity** | **Very important** | 27 | (24.77%) | 11 | (15.71%) | 38 | (21.23%) |
| | **Important** | 78 | (71.56%) | 53 | (75.71%) | 131 | (73.18%) |
| | **Not important** | 4 | (3.67%) | 6 | (8.57%) | 10 | (5.59%) |
| **Course duration** | **Less than 30 minutes** | 34 | (31.19%) | 12 | (17.39%) | 46 | (25.84%) |
| | **30 to 60 minutes** | 45 | (41.28%) | 32 | (46.38%) | 77 | (43.26%) |
| | **More than 60 minutes to 90 minutes** | 25 | (22.94%) | 21 | (30.43%) | 46 | (25.84%) |
| | **More than 90 minutes to 120 minutes** | 4 | (3.67%) | 3 | (4.35%) | 7 | (3.93%) |
| | **More than 120 minutes** | 1 | (0.92%) | 1 | (1.45%) | 2 | (1.12%) |
| **Course participation** | **Will participate in course** | 80 | (73.39%) | 33 | (47.14%) | 113 | (63.13%) |
| | **Will not participate in course** | 2 | (1.83%) | 11 | (15.71%) | 13 | (7.26%) |
| | **Uncertain to participate in course** | 27 | (24.77%) | 26 | (37.14%) | 53 | (29.61%) |

*Few participants did not complete all the questions.

has been described in areas such as tuberculosis, infection control, and antibiotic prescribing [21–25]. Similar to our study, the two tuberculosis (TB) studies revealed significant knowledge gaps among healthcare professions [23,25]. Interestingly, the Sohrabi et al study found that the majority of the participants could not correctly answer questions about TB treatment, but performed better in TB diagnosis and screening. Although participants in our study also performed poorly in management of pythiosis, the lowest performing areas were in laboratory investigation, epidemiology, and symptoms of vascular pythiosis. This could be because TB exposure in the educational and clinical environment is much more common than pythiosis, and a major focus of TB education is on screening and diagnosis to prevent TB spread. Moreover, the tuberculosis learning needs assessment in Iran showed that the subjective assessment from participant attitude did not correlate with the objective test assessment [25]. As we performed in our study, it is more precise to conduct an objective test in a learning needs assessment rather than subjective measures of competency. Consistently, the learning needs assessments in antibiotic prescribing and infection control studies identified practice patterns and knowledge gaps of clinicians, which helped inform professional societies of future educational efforts [21,22,24]. Similar to our study design, we will be using information gathered from our needs assessment to advise the creation of a pythiosis CPD program.

With respect to the other aspects of CRISIS and CPD planning, it is critical to determine learner preferences with respect to CPD content delivery in order to enhance engagement with the material [26]. Our study revealed that the course should be a mixture of synchronous and asynchronous online sessions. These preferences are consistent with previous studies that online learning is important. A study of emergency medicine physicians in the US showed that 65.62% of respondents would like to study via video, 13.98% via classroom (instructor-led training), and 5.68% via live webinars [27]. Additionally, a convergence mixed method study in Rwanda showed that participants preferred blended learning the most (online and face-to-face learning combined) [28]. This minor difference might be from the varied learning style preferences in other countries, in addition to challenges for delivering online CPD content in some areas of the world.

With respect to building a pythiosis CPD program, it is important to note that early diagnosis is crucial in decreasing morbidity and mortality, and the time to diagnosis and surgery are several factors associated with pythiosis outcomes [6–8]. In Thailand, the time from the first medical encounter to radical surgery ranged from 1–60 days [6]. In our pythiosis needs assessment, we identified three areas with the lowest percentage correct: laboratory investigation, epidemiology, and symptoms of vascular pythiosis. The low scores could be explained by the lack of exposure in each domain, attributed to the disease rarity. These three areas are critical to establish a diagnosis of pythiosis. For example, knowing which labs to order, how to interpret results, and how to transport specimens (laboratory investigations); understanding the clinical presentation of pythiosis (symptomatology); and appreciating that pythiosis is most prevalent in India and Thailand (epidemiology) is important to establish a diagnosis early in the disease process. By creating a pythiosis CPD program that focuses on content related to diagnosis, we will be able to address this knowledge gap with hopes of decreasing pythiosis morbidity and mortality. Regarding content delivery, the residents prefer online training courses with a mixture of synchronous and asynchronous sessions, with the course being less than 90 minutes in duration.

This is the first study in developing a learning needs assessment for vascular pythiosis and has many strengths. First, the instruments in this study were constructed by experts in Mycology, Epidemiology, and Medical Education Research Group (MERG), validated by four experts in both infectious disease and medical education, and adhered to survey development guidelines [17]. Second, this study included an objective assessment in addition to a subjective one about learner preferences and attitudes toward the topic. Third, learners' preferences in CPD followed the CRISIS acronym which provides a holistic view for CPD development. However, there are also limitations in this study. First, the participants in this study were internal medicine and surgery residents at a single institution due to feasibility. This selection does not encompass all physicians in Thailand, particularly the general practitioners who serve as frontline healthcare providers. Nevertheless, based on this study, we can infer that there may be a knowledge gap among general practitioners as well. Second, this study focused on training courses, which is one type of CPD delivery method, but it does not cover all delivery strategies. Additionally, the sampling technique was voluntary sampling–it is possible that residents with prior knowledge or interest in pythiosis were more likely to participate in the study, which may have resulted in an over-estimation of pythiosis knowledge in the study group. Moreover, the varying proportion of senior and junior residents might affect the performance outcomes. Further studies should investigate a broader range of participants including general practitioners, family medicine clinicians, and community clinicians, other CPD delivery strategies, increasing awareness of pythiosis (particularly in high-risk populations), and the development and evaluation of a pythiosis CPD program.

## Conclusion

The learning needs assessment revealed a knowledge gap in vascular pythiosis in internal medicine and surgery residents who encounter this deadly disease, specifically related to pythiosis diagnosis (laboratory investigation, symptoms of pythiosis, and epidemiology). Therefore, the pythiosis CPD program will focus on these areas, in addition to other important areas to pythiosis management, with hopes of closing this knowledge gap and improving patient survival via early detection with prompt surgical management. Moreover, the pythiosis CPD program should be an online synchronous and asynchronous format based on learners' preferences.

## Supporting information

**S1 Supplement. Supplement 1.** Survey for evaluating the necessity of developing the vascular pythiosis training program.
(DOCX)

**S2 Supplement. Supplement 2.** Table of specification of test items.
(DOCX)

**S1 Data. Survey record form in King Chulalongkorn Memorial Hospital.**
(XLSX)

## Acknowledgments

We extend our sincere gratitude to Miss Atthanee Jeeyapant for her assistance with statistical analyses in this study.

## Author Contributions

**Conceptualization:** Surachai Leksuwankun, Rongpong Plongla, Nitipong Permpalung, Saman Nematollahi.

**Data curation:** Surachai Leksuwankun, Rongpong Plongla, Saman Nematollahi.

**Formal analysis:** Surachai Leksuwankun.

**Funding acquisition:** Pattama Torvorapanit.

**Investigation:** Surachai Leksuwankun, Nathanich Eamrurksiri, Pattama Torvorapanit, Kasidis Phongkhun, Tanaporn Meejun, Karan Srisurapanont, Jaedvara Thanakitcharu, Bhoowit Lerttiendamrong, Achitpol Thongkam, Kasama Manothummetha, Nipat Chuleerarux, Chatphatai Moonla, Saman Nematollahi.

**Methodology:** Surachai Leksuwankun, Nitipong Permpalung, Saman Nematollahi.

**Project administration:** Nattapong Langsiri.

**Resources:** Navaporn Worasilchai, Ariya Chindamporn.

**Supervision:** Rongpong Plongla, Nitipong Permpalung, Saman Nematollahi.

**Validation:** Surachai Leksuwankun, Rongpong Plongla.

**Visualization:** Surachai Leksuwankun.

**Writing – original draft:** Surachai Leksuwankun, Saman Nematollahi.

**Writing – review & editing:** Surachai Leksuwankun, Rongpong Plongla, Nathanich Eamrurk-siri, Pattama Torvorapanit, Kasidis Phongkhun, Nattapong Langsiri, Tanaporn Meejun, Karan Srisurapanont, Jaedvara Thanakitcharu, Bhoowit Lerttiendamrong, Achitpol Thongkam, Kasama Manothummetha, Nipat Chuleerarux, Chatphatai Moonla, Navaporn Worasilchai, Ariya Chindamporn, Nitipong Permpalung, Saman Nematollahi.

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
