## [Decision Letter · Decision Letter 0]

11 Jan 2024

Dear Dr. Plongla,

Thank you very much for submitting your manuscript "Needs Assessment of a Pythiosis Continuing Professional Development Program" for consideration at PLOS Neglected Tropical Diseases. As with all papers reviewed by the journal, your manuscript was reviewed by members of the editorial board and by several independent reviewers. The reviewers appreciated the attention to an important topic. Based on the reviews, we are likely to accept this manuscript for publication, providing that you modify the manuscript according to the review recommendations. 

Sincerely,

Joshua Nosanchuk, MD

Section Editor

Reviewer's Responses to Questions

**Key Review Criteria Required for Acceptance?**

**Methods**

-Are the objectives of the study clearly articulated with a clear testable hypothesis stated?

-Is the study design appropriate to address the stated objectives?

-Is the population clearly described and appropriate for the hypothesis being tested?

-Is the sample size sufficient to ensure adequate power to address the hypothesis being tested?

-Were correct statistical analysis used to support conclusions?

-Are there concerns about ethical or regulatory requirements being met?

Reviewer #1: Methodology, populations, sample size, and statistics were all clearly established. There is no concern about the ethical issue.

Reviewer #2: (No Response)

Reviewer #3: The authors conducted a needs assessment survey and obtained a good survey completion rate of 67%. The study objectives were clearly articulated and was appropriate to address stated objectives. Based on the author's statistical analysis it appears the sample size was adequate. The sampling technique was voluntary sampling – it’s possible that residents with some knowledge of or interest in pythiosis were more likely to take the survey which might have resulted in an over-estimation of knowledge of pythiosis in the sample group.

**Results**

-Does the analysis presented match the analysis plan?

-Are the results clearly and completely presented?

-Are the figures (Tables, Images) of sufficient quality for clarity?

Reviewer #1: Everything seems OK to follow.

Reviewer #2: (No Response)

Reviewer #3: Analysis presented matches analysis plan and results were clearly presented. Tables and figures were appropriate and easy to understand.

**Conclusions**

-Are the conclusions supported by the data presented?

-Are the limitations of analysis clearly described?

-Do the authors discuss how these data can be helpful to advance our understanding of the topic under study?

-Is public health relevance addressed?

Reviewer #1: Everything seems OK to follow.

Reviewer #2: (No Response)

Reviewer #3: The author's conclusions are appropriate given the survey findings. An educational program as proposed by the authors would likely only be applicable in regions such as South and Southeast Asia as the disease is extremely uncommon in the rest of the world. It could be generalizable though to infectious disease specialists even outside of these regions.

**Editorial and Data Presentation Modifications?**

Reviewer #1: (No Response)

Reviewer #2: (No Response)

Reviewer #3: Recommended minor revisions:

Line 133-134 – assertion that all vascular pythiosis cases are reported from Thailand is incorrect and needs to be corrected. See Travel Med Infect Dis. 2022 Jul-Aug:48:102349. by Perkins et al. in which two human cases of vascular pythiosis from North America are reported (one acquired in Jamaica, one acquired in Texas).

Line 328 – Pythiosis occurs outside of India and Thailand. Authors should change this sentence to something like “pythiosis is most prevalent in India and Thailand.”

Suggestions:

Line 128 – would change “low prevalent” to low-prevalence 

Line 291 – in the U.S. we usually use the term “infection control” instead of “infectious control” and the authors could consider changing this. 

Line 304 – see above comment

**Summary and General Comments**

Reviewer #1: Leksuwankun et al. conducted a questionnaire-based research to survey knowledge of pythiosis in Thailand among residency trainees in King Chulalongkorn Memorial Hospital. A small sample size limited the study, and some issues need to be addressed point-by-point as follows:

- The authors stated that vascular pythiosis is uncommon and challenging to diagnose due to similar clinical presentations to other fungal infections such as mucormycosis, talaromycosis, and aspergillosis. This infection is usually underdiagnosed due to several factors, such as slow progress or gradual onset of vascular invasion, patient and physician under-recognition, and rarity of the disease.

- Primary physicians, family doctors, or community doctors are additional focused groups to develop CPD to enhance early diagnosis of pythiosis and referral. According to the rarity but high disease burdens, early recognition and diagnosis should effectively reduce this threat. 

- Proportions of PGY from Y1-Y5 were too different; this may affect the performance assessment. 

- TB exposure in the educational program is much more common in Thailand. Meanwhile, in the case of rare diseases, other examples of developing physicians' diagnosis performance, such as infrequent diseases, should be reviewed and discussed. 

- What are the root causes of underscore in vascular pythiosis lab investigations, symptoms, and epidemiology among residents? This point is intriguing to discuss.

- Another future research and development would be increasing awareness of this invasive disease, particularly in high-risk populations.

Reviewer #2: (No Response)

Reviewer #3: This paper addresses an important aspect this this rare infection – lack of knowledge of pythiosis by health care providers. As mentioned in the paper, time to diagnosis is an important factor in the efficacy of subsequent therapy so an educational program, such as that suggested by the authors, could be very important in improving outcomes of patients with this rare disease. Education of providers on the subject of pythiosis has rarely, if ever, been addressed in the medical literature and this paper is an important contribution to the literature on pythiosis.

PLOS authors have the option to publish the peer review history of their article (what does this mean?). If published, this will include your full peer review and any attached files.

Reviewer #1: No

Reviewer #2: No

Reviewer #3: No

Figure Files:

Data Requirements:

Reproducibility:

References

---

## [Editor Report · Decision Letter 1]

16 Feb 2024

Dear Dr. Plongla,

We are pleased to inform you that your manuscript 'Needs Assessment of a Pythiosis Continuing Professional Development Program' has been provisionally accepted for publication in PLOS Neglected Tropical Diseases.

Best regards,

Joshua Nosanchuk, MD

Section Editor

Joshua Nosanchuk

Section Editor

The authors are to be commended for their robust response to reviewer comments.

---

## [Editor Report · Acceptance letter]

21 Feb 2024

Dear Dr. Plongla,

We are delighted to inform you that your manuscript, "Needs Assessment of a Pythiosis Continuing Professional Development Program," has been formally accepted for publication in PLOS Neglected Tropical Diseases.

Best regards,

Shaden Kamhawi

co-Editor-in-Chief

Paul Brindley

co-Editor-in-Chief
